# Use of Glutathione, Pure or as a Specific Inactivated Yeast, as an Alternative to Sulphur Dioxide for Protecting White Grape Must from Browning [note 1]

**DOI:** 10.3390/foods13020310

**Published:** 2024-01-18

**Authors:** Marco Bustamante, Pol Giménez, Arnau Just-Borràs, Ignasi Solé-Clua, Jordi Gombau, José M. Heras, Nathalie Sieczkowski, Mariona Gil, José Pérez-Navarro, Sergio Gómez-Alonso, Joan Miquel Canals, Fernando Zamora

**Affiliations:** 1Departament de Bioquímica i Biotecnologia, Facultat d’Enologia de Tarragona, Universitat Rovira i Virgili, C/Marcel.li Domingo 1, 43007 Tarragona, Spain; marcoandres.bustamante@urv.cat (M.B.); pol.gimenez.gil@gmail.com (P.G.); jmcanals@urv.cat (J.M.C.); 2Lallemand Bio S.L., C/Galileu 303, 1ª Planta, 08028 Barcelona, Spain; 3Instituto de Ciencias Aplicadas, Facultad de Ingeniería, Campus Providencia, Universidad Autónoma de Chile, Sede Santiago, Av. Pedro de Valdivia 425, Providencia, Santiago 7510552, Chile; mariona.gil@uautonoma.cl; 4Instituto Regional de Investigación Científica Aplicada, Universidad de Castilla-La Mancha, 13001 Ciudad Real, Spain

**Keywords:** grape must, glutathione, inactivated dry yeasts, browning, laccase

## Abstract

One of the problems that most seriously affects oenology today is enzymatic browning, especially when grapes are infected by grey rot. We studied the capacity of glutathione (GSH) and a specific inactivated dry yeast rich in glutathione (IDY-GSH) to protect white grape must from browning compared to that of sulphur dioxide (SO_2_). The results indicate that SO_2_ drastically reduces the oxygen consumption rate (by around 72%), protects hydroxycinnamic acids from oxidation and prevents grape must against browning even in the presence of laccase. Specifically, the presence of SO_2_ reduced the colour’s blue–yellow component (*b**) by around 91% in control conditions and around 76% in the presence of laccase. GSH, pure or in the form of IDY-GSH, also reduces the oxygen consumption rate (by 23% and 36%, respectively) but to a lesser extent than SO_2_. GSH also favours the formation of grape reaction product (GRP) from hydroxycinnamic acids and effectively protects grape must against browning in healthy grape conditions. Specifically, the presence of GSH reduced *b** by around 81% in control conditions. Nevertheless, in the presence of laccase, it was not effective enough, reducing *b** by around 39% in the case of pure GSH and 24% in the case of IDY-GSH. Therefore, both forms of GSH can be considered as interesting alternative tools to SO_2_ for preventing browning in white grape must, but only when the grapes are healthy.

## 1. Introduction

One of the major problems in oenology currently is enzymatic browning [1], especially when grapes are affected by the filamentous fungi *Botrytis cinerea* [2]. Browning is an enzymatic oxidation process in several foods that causes the appearance of a brown colour, which sometimes leads consumers to reject them [3]. This problem is particularly damaging in oenology since grape must is very susceptible to browning [4].

Two enzymes belonging to the broad family of oxidoreductases called polyphenol oxidases (PPOs) [5] are responsible for enzymatic browning: tyrosinase (EC 1.14.18.1), which is naturally present in grape berries [6], and laccase (EC 1.10.3.2), which is only present in grapes infected by epiphytic fungi, mainly *Botrytis cinerea* [7].

Both enzymes are multicopper oxidases that use oxygen to primarily oxidize some phenolic compounds, such as hydroxycinnamic acids and other diphenols [8]. Due to the polyphenol oxidase activity, some diphenols are oxidized to diquinones, which polymerize later to form melanins [3]. These melanins are responsible for the darkening of the yellow colour in white wines (browning) and for the loss of the red colour in red wines (oxidasic haze). When grapes are affected by grey rot, the laccase activity can be much greater than that of tyrosinase in healthy grapes [9], and therefore the risk of browning is much higher. Moreover, laccase can oxidize a greater range of substrates than tyrosinase [4,7].

The most common method used in wineries to protect grape must from browning is adding sulphur dioxide because it is a powerful inhibitor of tyrosinase and laccase [10]. Sulphur dioxide is also used in winemaking for its antimicrobial properties [11]. Giménez et al. [8] reported in a kinetic study that sulphur dioxide drastically reduces the Vmax and increases the K0.5 of laccase, which shows that this additive not only inhibits this enzyme but also decreases its affinity versus the substrates. Tyrosinase is very sensitive to sulphur dioxide; therefore, small doses of this additive inactivate it. However, laccase is more resistant to sulphur dioxide and, unlike tyrosinase, it can be present in wine after alcoholic fermentation [10,12].

However, the recent tendency in winemaking is to diminish and even eliminate sulphur dioxide owing to its negative effects on the environment [13] and human health [14]. Therefore, some other additives have been proposed as alternatives to sulphur dioxide to prevent browning, such as ascorbic acid [15], inert gases [16], oenological tannins [17] and, more recently, the use of reduced glutathione [8,18,19] or inactivated dry yeasts rich in glutathione [20,21].

Ascorbic acid consumes oxygen very quickly [17,22], making it unavailable to phenol oxidases. However, during the oxidation of ascorbic acid, oxygen is reduced to hydrogen peroxide, which can cause subsequent oxidation in the wine [4]. Therefore, the use of ascorbic acid always requires the presence of sulphur dioxide to prevent wine oxidation [23]; it must be considered as a complement and never as an alternative.

Using inert gases minimizes the presence of oxygen, and by eliminating its substrate, oxidation caused by phenol oxidases is avoided [12,15].

Oenological tannins have been shown to be effective inhibitors of laccase activity [17]. Consequently, when musts and wines are supplemented with oenological tannins, they exert a protective effect on the colour of the wine.

Glutathione reduces browning because it reacts with the ortho-quinones originating from the enzymatic oxidation of ortho-diphenols to produce 2-S-glutathionylcaftaric acid, commonly known as the grape reaction product (GRP). In this way, glutathione traps the ortho-quinones in a colourless form and thus limits the formation of melanins [18,24].

Among all these possible alternatives to sulphur dioxide for preventing browning, glutathione is probably the most promising. The use of glutathione in grape must and wine was authorized in 2015 by the International Organization of Vine and Wine (OIV) at a maximum dosage of 20 mg/L [25,26]. This dose is considered to be very safe from a health point of view, given that the maximum daily intake considered by some food health agencies is 50 mg/day [27,28]. Nevertheless, glutathione is not often used in its pure form due to its very high price. Therefore, using certain inactivated dry yeasts as a source of glutathione (IDY-GSH) has been much more successful, as it is a more economical solution [20,21]. Using these inactivated dry yeasts with guaranteed levels of glutathione in winemaking was authorized by the OIV in 2018 [29].

In a recent article [30], our research group studied the influence of glutathione on the oxygen consumption kinetics and colour of grape must under similar conditions as those described in this work. Since the results were very interesting, we decided to repeat the experience, analysing hydroxycinnamic acids and the GRP as well in order to try to better understand the mechanism by which glutathione protects the must from browning. The aim of this paper was therefore to study the mechanism by which reduced glutathione, whether pure or as commercial inactivated dry yeasts rich in glutathione, exerts a protective effect against the oxidation caused by polyphenol oxidases.

## 2. Materials and Methods

### 2.1. Chemicals and Equipment

Syringaldazine (purity ≥ 98%), polyvinylpolypyrrolidone (PVPP) (purity ≥ 98%) and FeSO_4_·7H_2_O (purity ≥ 99%) were purchased from Sigma-Aldrich (Madrid, Spain). L-(·+)-tartaric acid (purity ≥ 99.5%), sodium hydroxide (purity ≥ 98%), sodium acetate (purity ≥ 99%), acetonitrile (purity ≥ 99%), methanol (purity ≥ 99%) and CuSO_4_·5H_2_O (purity ≥ 99%) were purchased from Panreac (Barcelona, Spain). Ethanol (96% vol.) was supplied by Fisher Scientific (Madrid, Spain). L-glutathione reduced (purity ≥ 98%) was purchased from Sigma-Aldrich (Madrid, Spain). *trans*-Caftaric acid (purity ≥ 95%) was purchased from Phytolab (Vestenbergsgreuth, Germany). Cellulose membranes of 3.5 KDa (6.4 mL/cm) were supplied by Spectrum Laboratories, Inc. (Rancho Dominguez, CA, USA). The equipment used was as follows: an Hpand and Entris II Series Analytical Balance (Sartorius, Goettingen, Germany), a UV-Vis Helios Alpha ™ spectrophotometer (Thermo Fisher Scientific Inc., Waltham, MA, USA), a Heraeus™ Primo™ centrifuge (Thermo Fisher Scientific Inc., Waltham, MA, USA), and an Agilent Series 1200 liquid chromatograph (Agilent, Germany) equipped with a photodiode array detector (G1315D) and a Zorbax Eclipse XDB C18 column (4.6 × 150 mm).

### 2.2. Collecting and Preparing the Samples

Grapes from *Vitis vinifera* cv. Muscat of Alexandria grafted onto 110 Richter rootstock (planted in 2014) were obtained from the experimental vineyard of the Oenology Faculty of the Rovira i Virgili University in Constantí (AOC Tarragona; 41°08′44.1″ N and 1°11′51.0″ E) during the 2022 vintage. The grape vines were grown on a limestone clay soil 59 m above sea level, and groundwater was located at a depth of around 4 m. The vines were trained on a vertical trellis system and arranged in rows 2.80 m apart, with 1.20 m of spacing within each row. They were pruned following a double Cordon de Royat system, with 16 buds per vine remaining after winter pruning. The vineyard was managed according to standard viticultural practices for the cultivar and region, and chemical treatments were applied for prevention of oidium (*Uncinula necator*), powdery mildew (*Plasmopara viticola*), grapevine moth (*Lobesia botrana*) and grey rot (*Botrytis cinerea*). The weather conditions of the vintage were as follows: annual rainfall: 355 mm; and average temperature: 16.7 °C. More specifically, the weather conditions from budburst to harvest were as follows: rainfall: 183 mm; and average temperature: 22.6 °C. The grapes were harvested manually when they were considered to be sufficiently ripe (with potential ethanol content around 12.5% and tartaric acid/L titratable acidity of 6.0 g).

The grapes were pressed in an environment saturated with nitrogen with a vertical manual press until 60% of their must was extracted and transferred into a 750 mL bottle, which was also saturated with nitrogen.

### 2.3. Synthetic Buffer

A solution of 4 g/L of L-(+)-tartaric acid, 3 mg/L of iron in the form of FeCl_3_·6H_2_O and 0.3 mg/L of copper in the form of CuSO_4_·5H_2_O adjusted to pH of 3.5 with NaOH was used for all experiments.

### 2.4. Production of the Laccase Enzyme and Measurement of Enzymatic Activity

Active laccase extracts were obtained from the 213 isolate strain of *Botrytis cinerea* following the procedure reported by Vignault et al. [17]. This laccase extract was treated with 0.16 g/mL of PVPP for 10 min and centrifuged at 7500 rpm for 10 min. The supernatant was subsequently dialyzed with 3.5 KDa cellulose membrane for two days against a 0.3 M ammonium formate solution and for two more days against distilled water. The laccase activity was determined using the syringaldazine test method [31].

### 2.5. Measurement of Oxygen Consumption Kinetics

The experimental conditions were those described by Giménez et al. [30]. The assays were carried out in clear glass flasks (66 mL) with an oxygen sensor spot (PreSens Precision Sensing GmbH, order code: SP-PSt3-NAU-D5-CAF; batch number: 1203-01_PSt3-0828-01, Regensburg, Germany) for noninvasively measuring the dissolved oxygen via luminescence (Nomasense TM O2 Trace Oxygen Analyzer, Nomacorc S.A., Thimister Clermont, Belgium).

The grape must was diluted (20% (*v*/*v*)) with the synthetic buffer mentioned above to reduce the oxygen consumption rate of the grape must so it could be monitored with greater precision and to uniformize the pH of the different measurements. More specifically, 13 mL of grape must and 52 mL of buffer were added to each flask, with the different antioxidant agents. The antioxidants used were as follows: sulphur dioxide (20 mg/L in the form of potassium disulphite), pure reduced glutathione (20 mg/L) and a commercial inactivated dry yeast (Glutastar™, Lallemand Inc., Montreal, QC, Canada) rich in glutathione (400 mg/L). A control without anything added to it was also prepared. These assays were also performed by combining sulphur dioxide with each of the other antioxidant agents with and without the addition of 2 U/mL of laccase (enzyme units or μmol/minute). All assays were performed in triplicate.

The bottles were immediately saturated in oxygen (around 7–8 mg/L of O_2_) by manual vigorous stirring for a few seconds. The oxygen concentration was measured [32] periodically until asymptotic behaviour was reached (around 3 h) to determine the oxygen consumption kinetics. All measurements were performed at 22 ± 2 °C. The total oxygen consumption capacity (TOCC) was determined using a mathematical model previously described by Pons-Mercadé et al. [33]. When the oxygen concentrations were below 1 mg/L or oxygen consumption attained an asymptotic behaviour, 50 mg/L of SO_2_ was added to the samples to avoid the colour’s evolution. The samples were then used for colour measurements and for HPLC analysis of hydroxycinnamic acids and GRP.

### 2.6. Colour Measurements

The intensity of the yellow colour (A420nm) and the Cie*L***a***b** coordinates of the samples were measured according to Ayala et al. [34]. The data were processed using the MSCV software [35]. The total colour difference (ΔE*ab**) was calculated using the following formula: ΔE*ab** = ((*L**_1_ − *L**_2_)^2^ + (*a**_1_ − *a**_2_)^2^ + (*b**_1_ − *b**_2_)^2^)^1/2^, where *L** is the lightness; *a** is the colour’s green–red component; and *b** is the colour’s blue–yellow component. ΔE*ab** is used to determine whether the difference between two samples can be detected visually by the human eye. Generally, it is considered that the difference is visible to the human eye when ΔE*ab** > 3 units [36,37].

### 2.7. Hydroxycinnamic Acid and GRP Analysis via HPLC-DAD

Hydroxycinnamic acids and GRP were analysed by reversed-phase HPLC−DAD following the method described by Lago-Vanzela et al. [38]. This comprised an Agilent Series 1200 HPLC (Agilent, Waldbronn, Germany) system equipped with a DAD (G1315D) coupled to an Agilent Chem Station (version B.01.03) data processing station. The samples were filtered (0.20 μm, polyester membrane, Chromafil PET 20/25, Machery-Nagel, Düren, Germany) and then injected (20 μL) into a Zorbax Eclipse XDB C18 column (4.6 × 150 mm). The solvent was composed of A (water/formic acid/acetonitrile (88.5:8.5:3, *v*/*v*/*v*)), B (water/formic acid/acetonitrile (41.5:8.5:50, *v*/*v*/*v*)) and C (water/formic acid/methanol (1.5:8.5:90, *v*/*v*/*v*)), and the flow rate was 0.19 mL/min. The gradient was as follows: from 0 to 37 min: A = 96%/B = 4%; from 37 to 51 min: A = 70%/B = 17%/C = 13%; from 51 to 57 min: A = 50%/B = 30%/C = 20%; and from 57 to 64 min A = 0%/B = 50%/C = 50%. The compounds were quantified by measuring the absorbance at 320 nm with the external calibration curve of caftaric acid. The different compounds were identified according to the retention time that was previously identified with an HPLC–DAD-ESI-MS/MS analysis using the same chromatographic conditions and system, coupled to a mass spectrometry detector with an electrospray ionization source (LC/MSD Trap VL, model G2445C VL). The mass spectrometer was operated under the following conditions: negative ionization mode; scan range OF 100–1000 *m*/*z*; N2 dry gas with flow of 8 L min^−1^; drying temperature of 350 °C; nebulizer at 40 psi; capillary at +3500 V; capillary exit offset of −68 V; skimmer 1 of −20 V; and skimmer 2 of −60 V. The identification of these compounds was based on their UV-Vis and MS/MS spectra, obtained from standards and those reported in the literature. Caftaric acid was used as calibration standard for quantification of all hydroxycinnamic acids and GRP. Table 1 shows the retention times and the molecular and fragment ions (*m*/*z*) obtained with negative ionization.

### 2.8. Statistics

All data are expressed as the arithmetic average ± standard deviation of three replicates. One-factor analysis of variance (ANOVA F-test) was carried out using the SPSS 15.0 software (SPSS Inc., Chicago, IL, USA) [39]. Significant differences between TOC, hydroxycinnamic acids, GRP, A420nm and *b** of the different samples were considered when the *p* value was less than 0.05.

## 3. Results

### 3.1. Oxygen Consumption Kinetics

Figure 1A shows the oxygen consumption kinetics of the diluted grape must either supplemented or not with sulphur dioxide, glutathione and a combination of both additives in the absence or presence of 2 U/mL of laccase.

The control diluted grape must without any additives initially consumed oxygen very quickly and then moderately until reaching values of around 2 mg/L of O_2_ after 3 h. This behaviour is probably due to the depletion of substrates for polyphenol oxidase, which slows down the reaction rate. In contrast, when the sample was supplemented with sulphur dioxide, the oxygen consumption rate (OCR) decreased drastically, confirming that this additive is a powerful inhibitor of the polyphenol oxidase (tyrosinase) present in the must of healthy grapes [10]. Supplementation with glutathione (GSH) also reduced the OCR but to a lesser extent than sulphur dioxide. This is probably because GSH reacts with the ortho-quinones formed by the action of polyphenol oxidases, and thus, their availability for oxidation is decreased [18,30]. Figure 1A also shows the OCR of all these samples in the presence of laccase. In general, laccase supplementation accelerated the OCR of the control and of the sample supplemented with GSH in comparison to that of their corresponding samples without laccase. In contrast, the samples supplemented with sulphur dioxide alone or in combination with GSH showed a similar behaviour to that of the corresponding samples without laccase. This confirms that sulphur dioxide is a very effective inhibitor not only of tyrosinase but also of laccase [8,11].

Figure 1B shows the OCR of the diluted grape must supplemented with inactivated dry yeasts rich in glutathione (IDY-GSH) in the absence and presence of laccase. This figure also shows the influence of sulphur dioxide and of the combination of the two treatments. In general, the observed behaviour for IDY-GSH in the absence of laccase was very similar to that reported for pure GSH. When laccase was present, the OCR of the control and especially that of IDY-GSH was also accelerated, whereas supplementation with sulphur dioxide reduced these a lot in a similar way to that in the absence of laccase.

Figure 1 gives the reader an idea of the OCR but cannot be used to establish statistically significant differences within the different experimental groups. Therefore, a previously reported kinetic model [33] was applied to statistically compare the results of this graph and calculate the total oxygen consumption (TOC) of the different samples. This model consists of representing the inverse of the consumed oxygen versus the inverse of the elapsed time. Using this mathematical model, the following equation can be established.
1/[O_2_] = A/t + B

In which O_2_ is the concentration of consumed oxygen (expressed in mg/L), A is the slope of the regression line, B is the point of intersection with the ordinate axis and t is the time (expressed in minutes). After clearing up the oxygen in the equation, the TOC can be calculated as the limit when the time proceeds towards infinity.

Figure 2A shows the calculated TOC of the diluted grape must supplemented or not with sulphur dioxide, GSH and a combination of the two additives in the absence or presence of 2 U/mL of laccase.

This graph confirms that the TOC of the samples supplemented with sulphur dioxide, alone or in combination with GSH, was significantly lower than that of their corresponding controls. Supplementation with GSH also significantly reduced the TOC but to a lesser extent than that of sulphur dioxide. No significant differences were found in the TOC between the samples with and without supplementation with laccase. This indicates that laccase increased the initial OCR but not the total consumption of oxygen.

The supplementation with IDY-GSH produced very similar results to that of GSH in the absence of laccase since it also reduced the TOC significantly in comparison with the control (Figure 2B). However, in the presence of laccase, IDY-GSH did not reduce the TOC, which indicates that in these conditions, it is not as effective as pure GSH for protecting grape must against oxidation.

### 3.2. Hydroxycinnamic Acids and GRP

Figure 3A shows the hydroxycinnamic acids and GRP concentrations of the original grape must and the samples supplemented or not with sulphur dioxide, GSH and the combination of the two additives after oxygen consumption.

In the absence of laccase, the original grape must was very rich in hydroxycinnamic acids, especially in caftaric acid, whereas the concentration of GRP was practically nil. These data confirm that the grape must was very well protected against oxidation. In the absence of sulphur dioxide, the hydroxycinnamic acid concentration of the control grape must drastically decreased after oxygen consumption without the appearance of GRP. This indicates that nearly all the hydroxycinnamic acids were oxidized to the corresponding ortho-quinones. The very low concentration of GRP indicates that the original grape must was very poor in GSH. In contrast, when the grape must was supplemented with sulphur dioxide, the hydroxycinnamic acid concentration remained at similar levels to those in the original grape must, which confirms that this additive really inhibits the action of tyrosinase. This result is in accordance with the observed reduction in the oxygen consumption caused by sulphur dioxide that was previously described (Figure 1 and Figure 2). In contrast, hydroxycinnamic acids completely disappeared when the grape must was supplemented with GSH in the absence of sulphur dioxide, and the concentration of GRP increased very significantly. It therefore appears that almost all hydroxycinnamic acids were oxidized to ortho-quinones, which were immediately converted to GRP upon their reaction with GSH. When GSH and sulphur dioxide were added together, some of the hydroxycinnamic acids were conserved and some GRP was formed. These results are more difficult to interpret since it seems that sulphur dioxide inhibited the action of tyrosinase but to a lesser extent than when only sulphur dioxide was added because some of the hydroxycinnamic acids had been transformed into GRP.

Very similar results were obtained when the samples were supplemented with 2 U of laccase. The only remarkable difference was that the levels of GRP of the samples supplemented with GSH were lower, probably because laccase, unlike tyrosinase, can oxidize GRP [18].

Figure 3B shows the hydroxycinnamic acids and GRP concentrations of the original grape must and the samples supplemented or not with sulphur dioxide, IDY-GSH and the combination of the two additives. In this case, the original grape must contained GRP, and the level of total hydroxycinnamic acids was somewhat lower than that in the previous experiment. These data indicate that the grape must underwent some oxidation since GRP is formed by adding GSH to the ortho-quinone formed by the oxidation of caftaric acid [18,24]. When no sulphur dioxide was added, the GRP concentration was slightly but significantly lower than that in the original grape must, and the concentration of hydroxycinnamic acids was very low, which would indicate that tyrosinase nearly completely oxidized these compounds. In contrast, when sulphur dioxide was present, the levels of GRP and hydroxycinnamic acids were similar to those of the original grape must, which indicates that this additive exerted a protective effect against oxidation. When the grape must was supplemented with IDY-GSH, the GRP concentration was similar to that of the original grape must, and the concentration of hydroxycinnamic acids was significantly lower than that in the original grape must but higher than that in the grape must without sulphur dioxide. Finally, the levels of GRP and hydroxycinnamic acids of the grape must supplemented with IDY-GSH and sulphur dioxide were very similar to those of the grape must supplemented only with sulphur dioxide. In general, in the presence of laccase, the levels of GRP and hydroxycinnamic acids were lower than in its absence, confirming that this polyphenol oxidase is very effective at oxidizing these compounds [4,7,9]. In fact, it has been reported that the content of hydroxycinnamic acids and flavonoids in grapes affected by Botrytis cinerea decreases in the comparison to healthy grapes [40,41].

### 3.3. Colour Parameters

Figure 4A shows the absorbance at 420 nm (A420nm) and the CIE*L** *a** *b** blue–yellow component (*b**) of the samples supplemented with sulphur dioxide, GSH and their combination in the absence and presence of laccase as indicators of the browning intensity.

The addition of SO_2_ exerted an evident protective effect against browning because the values of A420 and *b** were significantly lower than those in the control sample, confirming again that this additive effectively avoids this problem [4,10,11]. Supplementation with GSH, alone or in combination with sulphur dioxide, also significantly decreased A420 and *b** with respect to the control, confirming that this antioxidant really protects grape must against browning even in the absence of sulphur dioxide.

In the presence of laccase, although the samples supplemented with sulphur dioxide and GSH showed significantly lower values of A420 and *b** than the control, all of them had significantly higher values of the two parameters than their equivalent samples without laccase. These data confirm the high oxidative capacity of this polyphenol oxidase. It is also clear that it is more difficult to completely protect grapes affected by grey rot against browning, even when using sulphur dioxide.

Figure 4B shows the equivalent results of using IDY-GSH instead of pure GSH, and it can be seen that the results are very similar. Sulphur dioxide and IDY-GSH, alone or in combination with sulphur dioxide, exerted a significant protective effect against browning in the absence of laccase; however, this protection was not so effective when the samples were supplemented with this enzyme. It seems, therefore, that sulphur dioxide and also GSH, pure or in the form of IDY-GSH, can prevent must from browning in healthy grapes but their effect is not efficient enough when the grapes are affected by the presence of laccase.

Figure 5 shows the total colour difference (ΔE*ab**) between the different samples.

It is generally accepted that when the ΔE*ab** between two samples of wine is lower than three units, the human eye cannot distinguish between them [36,37]. The sample supplemented with sulphur dioxide was used as the reference to compare with all the other samples since this is the most common procedure for protecting grape must from browning. Consequently, when the ΔE*ab** between one sample and this control reference is greater or lower than three units, it indicates whether the browning intensity can or cannot be perceived by the human eye.

The results clearly show the protective effect of the various antioxidants against browning. Figure 5A shows the ΔE*ab** between the samples supplemented with GSH, alone or in combination with sulphur dioxide, and the reference supplemented only with sulphur dioxide. Under the conditions of healthy grape must (with the absence of laccase), the ΔE*ab** of the control sample was clearly higher than three units, and this value was much higher when laccase was present in the media. These data confirm that without sulphur dioxide, the grape must develops browning to a sufficient extent to be perceptible by the human eye. In contrast, supplementation with GSH, alone or in combination with sulphur dioxide, allowed the ΔE*ab** values to be kept below three units, which confirms that this antioxidant exerts a real protective effect.

However, when the samples were also supplemented with laccase, supplementation with GSH and even with sulphur dioxide was not effective enough to protect the samples against browning because the ΔE*ab** values were above three units.

When IDY-GSH was used instead of GSH (Figure 5B), the ΔE*ab** values showed a similar trend to those obtained for pure GSH (Figure 5A). IDY-GSH protected against browning (ΔE*ab** < three units) when no laccase was added; however, its protective effect was not enough when laccase was added (ΔE*ab** > three units).

Figure 6 shows pictures of the different samples at the end of the experiment.

In these pictures, it can be clearly perceived that the control sample without any addition underwent a clear browning process. In contrast, supplementation with sulphur dioxide, GSH, IDY-GSH and the combination of sulphur dioxide with these two last additives really protected the samples against browning in the absence of laccase since no visual differences in the browning intensity of the samples can be perceived by potential consumers. However, in the presence of laccase, the protective effects of GSH and IDY-GSH are not enough to prevent browning.

It seems therefore that, in light of these results, the use of glutathione, pure or in the form of inactivated dry yeasts, may be a useful tool to reduce or even replace the use of sulphur dioxide. It is clear, however, that the feasibility of its use in winemaking will also depend on its economic cost. A simple economic approach indicates that the use of sulphur dioxide at a normal dosage represents a cost of around 0.00077 USD/L, whereas the approximate cost of the use of pure glutathione is around 0.162 USD/L and that of inactivated dry yeasts rich in glutathione is around 0.010 USD/L. There is no doubt therefore that the cost of its use would be higher than that of sulphur dioxide. However, this increase in production cost might be considered acceptable, especially in the case of inactivated dry yeasts, given that wines with lower levels of or without sulphur dioxide have an increased value.

## 4. Conclusions

In this work, we analysed how supplementation with sulphur dioxide, glutathione and a specific inactivated dry yeast rich in glutathione influences the oxygen consumption kinetics, hydroxycinnamic acids and GRP concentrations, as well as the browning intensity of grape must under two conditions. The first condition was that the grape must was obtained from healthy grapes, while the second was the grape must was supplemented with the enzyme laccase to reproduce what happens when grapes are affected by grey rot. The results show that sulphur dioxide drastically reduces the oxygen consumption rate, protects hydroxycinnamic acids from oxidation and prevents grape must from browning, even in the presence of laccase. Supplementation with glutathione, pure or in the form of a specific inactivated dry yeast rich in glutathione, also reduces the oxygen consumption rate but to a lesser extent than sulphur dioxide. It favours the formation of GRP from hydroxycinnamic acids and effectively protects grape must against browning provided the grapes are healthy. However, in the presence of laccase, glutathione was not effective enough to prevent browning. It can be concluded, therefore, that glutathione and the specific inactivated dry yeast rich in glutathione can be considered adequate alternative tools to sulphur dioxide for preventing browning in white grape must obtained from healthy grapes.

## Figures and Tables

**Figure 1 foods-13-00310-f001:**
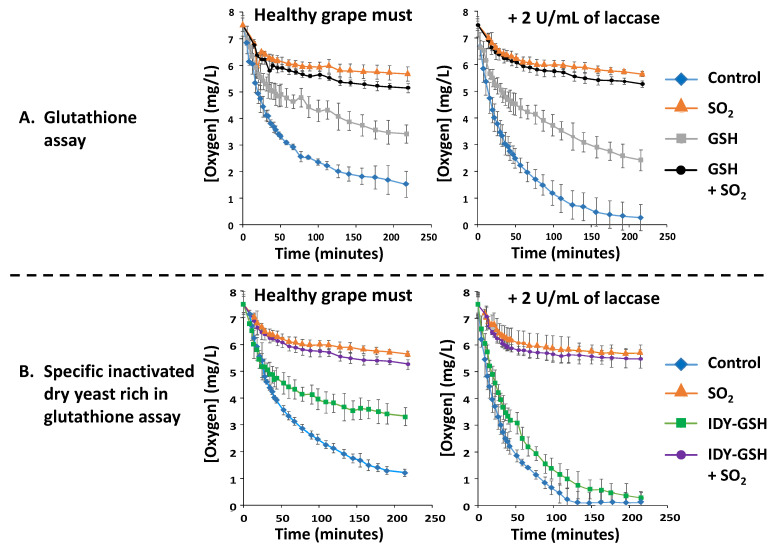
Oxygen consumption kinetics. Results are expressed as mean ± standard deviation of three replicates. GSH: Glutathione. IDY-GSH: Specific inactivated dry yeasts rich in glutathione.

**Figure 2 foods-13-00310-f002:**
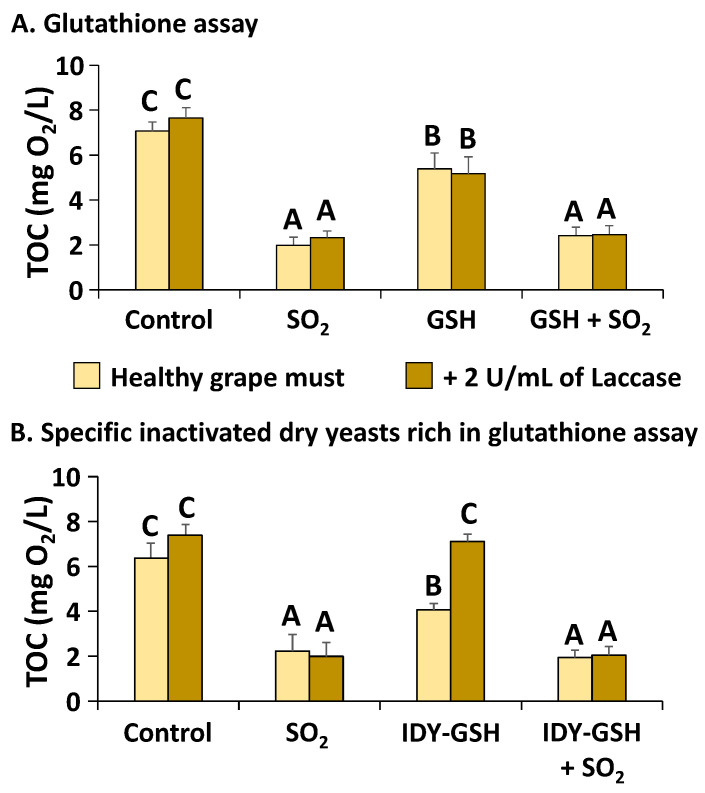
Total oxygen consumption (TOC). Results are expressed as mean ± standard deviation of three replicates. TOC: Total oxygen consumption. Glutathione; IDY-GSH: Specific inactivated dry yeasts rich in glutathione. Different letters indicate the existence of significant differences (*p* < 0.05).

**Figure 3 foods-13-00310-f003:**
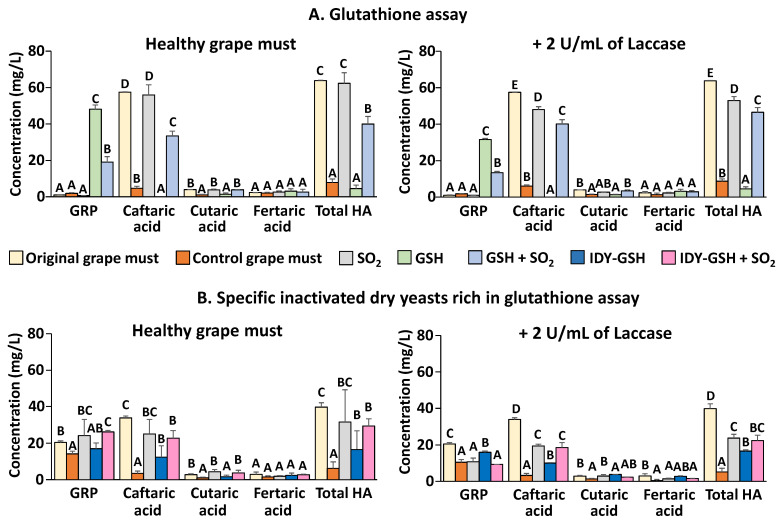
Hydroxycinnamic acids and GRP. Results are expressed as mean ± standard deviation of three replicates. GSH: Glutathione. IDY-GSH: Specific inactivated dry yeasts rich in glutathione. Different letters indicate the existence of significant differences (*p* < 0.05).

**Figure 4 foods-13-00310-f004:**
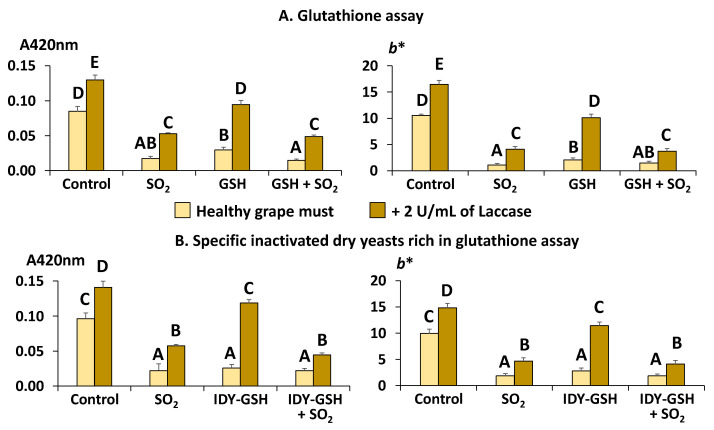
Browning intensity. Results are expressed as mean ± standard deviation of three replicates. A420nm: Absorbance at 420 nm. *b**: CIE*L***a***b** colour’s blue–yellow component. GSH: Glutathione. IDY-GSH: Specific inactivated dry yeasts rich in glutathione. Different letters indicate the existence of significant differences (*p* < 0.05).

**Figure 5 foods-13-00310-f005:**
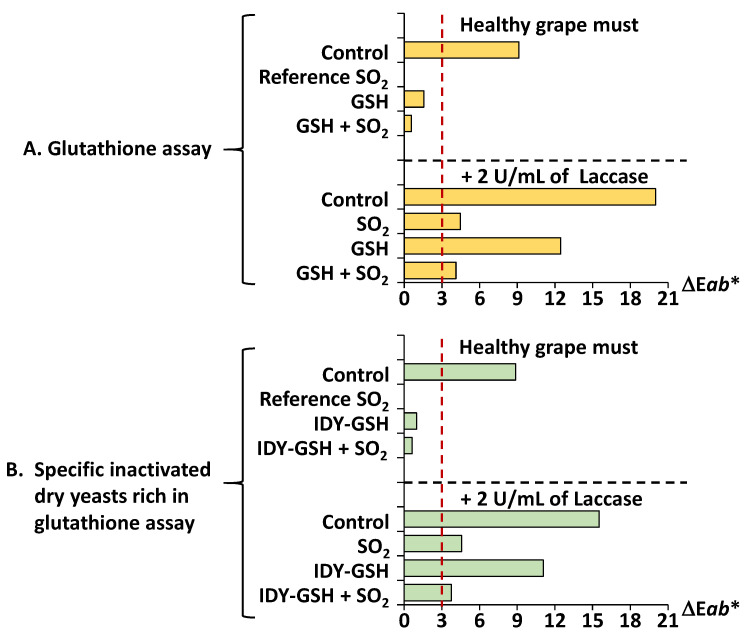
Total difference of colour (ΔE*ab**). Results are expressed as mean ± standard deviation of three replicates. GSH: Glutathione. IDY-GSH: Specific inactivated dry yeasts rich in glutathione. The red dotted line indicates the ΔE*ab** value below which the human eye cannot distinguish between two samples.

**Figure 6 foods-13-00310-f006:**
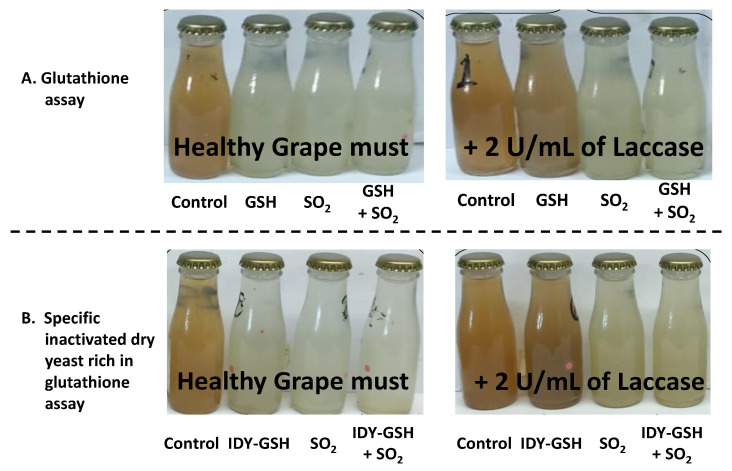
Influence of laccase, SO_2_, glutathione and a specific inactivated dry yeast rich in glutathione on the final colour of the samples. GSH: Glutathione. IDY-GSH: Specific inactivated dry yeasts rich in glutathione.

**Table 1 foods-13-00310-t001:** Retention times and molecular and fragment ions (*m*/*z*) obtained with negative ionization.

Compound	Retention Time	Molecular and Fragment Ions (*m*/*z*), Negative Ionization
GRP	3.24	616; 484; 272
Caftaric acid	3.81	311; 179; 149
Coutaric acid	5.51	295; 163; 149
Fertaric acid	7.81	325; 193; 149

## Data Availability

Data is contained within the article.

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
