# Peer review of "Use of Glutathione, Pure or as a Specific Inactivated Yeast, as an Alternative to Sulphur Dioxide for Protecting White Grape Must from Browning [Author-notes fn1-foods-13-00310]"

_foods, 2024, doi:10.3390/foods13020310_

Round 1

Reviewer 1 Report

Comments and Suggestions for Authors

Good idea and I found several reserchs were already published at the same point beagning from 2019 till now 

I have I few comments 

1. The abstract was very poor and full of repeating , it will be better if you rewrite it adding a background part about the problem you have

2. You should define the abbreviation in abstract 

3. All scientific names and L, a,b for color parameters should be write in italic font 

4. The introduction part also missing an important part about the nature of glutathione and the maximum dose can be used in food processing and its health effect or its toxicity 

5. You didn't show the type of yeast which rich in glutathione 

6.why you used only one concentration of glutathione or inactivated dry yeast rich with it? Depend on what?

7. The manuscript will be better if you write a small part about the feasibility study of using glutathione in wine processing 

Comments on the Quality of English Language

Moderate editing of English language required

Reviewer 2 Report

Comments and Suggestions for Authors

Very interesting work. Requires minor corrections.

1) Page 2, line 63

„However, ascorbic acid generates hydrogen peroxide after consuming oxygen and”

During the oxidation of ascorbic acid, oxygen is reduced to hydrogen peroxide.

2) line: 96, 97, 149, 221, 223

Please check the subscript in the chemical formulas.

3) Page 3, Line 101

Why all equipment is listed in the section: “2.1.Chemicals”?

4) Please standardize the units.

Page 3, Line 115:

“0.3 mg of copper/L”  0.3 mg/L of copper

Page 4, Line 140;  Figure 1, 3, 5, 6:

“2 U of laccase/mL” 2 U/mL of laccase

Page 4, Line 142:

“ 7–8 mg O2/L”  7–8 mg/L O2

Page 4, Line 148:

„50 mg of SO2/L” 50 mg/L of SO2

Figure 2:

„mg O2/L” mg/L O2

5) Page 3, Line 120:

PVPP not listed in chapter 2.1 (Chemicals)

6) Page 6, Line 223:

Oxygen concentration units are not provided.

7) Page 5, Figure 1:

Replacing dots with x's, crosses, asterisks, triangles or squares would increase the readability of the charts in the printed (monochromatic) version.

Reviewer 3 Report

Comments and Suggestions for Authors

Dear Authors,

This manuscript offers findings regarding application of different compounds which could be used instead of SO2 against must browning.

Abstract is too general. In the abstract are missing values of the most important results obtained in this study. Insert numbers in abstract.

In the section 2 are missing important information regarding the grape used in this experiment. Did you treat vineyard with any chemical substances for plant protection? What is type of soil on which was planted vineyard? Highlight climate condition during vegetative period of grape.

In the subsection 2.7. are missing important data regarding identification of hydroxycinnamic acid and GRP. What were retention times of compounds which were quantified? Insert it data.

In the line 174 you have mentioned that you were performed HPLC–DAD-ESI-MS/MS analysis. Insert the table in which will be highlighted all parameters (ionization mode ESI, MRM transition, cone voltage, collision energy and retention time) for determination of content of compounds. Did you use external standards? It is very important.

In the subsection 2.8. highlight which parameters from results were used in the statistical tests applied in this manuscript.

In the discussion highlight that content of specific phenolic compounds, among which are hydrohycinnamic acids and flavonoids in the grapes affected by Botrytis cinerea is decresed in the comparison with the healthy grapes. Kindly consider to cite Fermentation 9(7), (2023), 695.

Round 2

Reviewer 1 Report

Comments and Suggestions for Authors

Now all modifications  I asked have been done 

Comments on the Quality of English Language

 Minor editing of English language required

Reviewer 3 Report

Comments and Suggestions for Authors

Dear Authors, 

Thank you very much for revised version of manuscript and answers on questions. It is fine for me. Wish you all the best in the future work. 

Best regards, 
